# 'Moderate Islam' Made in the United Arab Emirates: Public Diplomacy and the Politics of Containment

**Panos Kourgiotis** 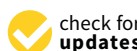

Department of Balkan, Slavic and Oriental Studies, University of Macedonia, GR-546 36 Thessaloniki, Greece; pkourgiotis@gmail.com

**Abstract:** This essay addresses the ideological utilization of religion in the international relations of the United Arab Emirates during the Arab Spring and beyond. By referring to the theoretical framework of public diplomacy and analyzing UAE regional and domestic attitudes, this essay intends to examine the politics of 'moderate Islam' in line with: (a) the monarchy's nation building visions for the 21st century; (b) its national rebranding strategies; (c) its geopolitical empowerment in the Gulf and the Middle East. Throughout our analysis, it is argued that even though 'moderate Islam' has been devised for creating 'soft power', it serves 'sharp power' as well. As will become obvious, this has been mainly the case as far as the containment of Political Islam is concerned.

**Keywords:** United Arab Emirates; Islam; public diplomacy; Gulf; Arab Spring; interfaith dialogue

## 1. Introduction

In the aftermath of the Arab Spring, certain countries in the region tried to manage the course of events in their bid to become regional superpowers. This has been especially true for the Gulf Cooperation Council (GCC) members, like Qatar or the United Arab Emirates, which is the country under question in this study (Young 2013).

Since their foundation in 1971, the Emirates have been a typical 'rentier state' (Issawi [1982] 2006, pp. 198–202; Noreng 2004, pp. 9–38) that uses its enormous resources as a means to legitimize the power sharing of several confederated dynasties; the most important among them being al-Nahyan and al-Maktoum dynasty, who govern Abu Dhabi and Dubai and hold the posts of presidency and premiership respectively (Foley 2002, pp. 38–40). The UAE were ranked, as of 2017, the 13th richest nation globally, holding the seventh and the sixth largest oil and natural gas reserves in the world (CIA The World Factbook 2017). Steps have been taken, though, to diversify the oil- and gas-dependent economy (Eno et al. 2016, pp. 102–4).

Regarding the monarchy's foreign relations, the excessive accumulation of wealth was manifested in active donor diplomacy. Billions of dollars have been paid so far in disbursements mainly to Africa, Afghanistan, the Arab countries and, to a lesser extent, the Americas, Europe and Oceania (Organization for Economic Co-Operation and Development 1971–2018). It is worth mentioning that for the years 2013–2014, the monarchy was the world's biggest donor of foreign aid, allocating 1.26 percent of its gross national income to official developmental aid (Embassy of the United Arab Emirates 2019a).

Nevertheless, donations are not the sole factor in shaping Emirati regional policies. In recent years, the country asserted a more aggressive, 'hard power' diplomacy that culminated in military interventions in a number of fronts, such as Bahrain, Syria, Libya and Yemen; still, regional hegemony cannot be sustained without the credibility of 'soft power' (Nye 2005). Until 2013, divergent priorities and security concerns over the political future of the Arab countries under transition, i.e., the role of the decades-old Islamist movements (Stein and Volpi 2015, pp. 286–88; Schwedler 2013, pp. 14–15), were crystallized into the emergence of two rival camps: the Saudi–Emirati axis and that of Qatar,

Turkey and the transnational Muslim Brotherhood (Bianco and Stansfield 2018, pp. 625–27). In this regard, post-Mohammad Morsi Egypt received 52 percent of the Emirates' total foreign aid budget for 2014 (MEE and Agencies 2015), against the backdrop of Abu Dhabi's public diplomacy initiatives, such as the so-called 'moderate Islam' campaign (Cafiero 2019).

Therefore, this essay intends to examine the utilization of Islam in the midst of Emirati regional adventurism and generous aid policies towards countries facing the threat of Political Islam or recovering from it. The study aims to show that promoting 'moderate Islam' far exceeds the scope of cultural/religious diplomacy; it is rather a by-product, of this small power's geopolitical transformation into an international actor that aspires to lead the Arab world (Carvalho-Pinto 2014, pp. 238–41). Setting money and coercive policies aside, the ultimate goal of this strategy is to create 'soft power', yet without sacrificing 'sharp power' tactics that stem from the fear of the Muslim Brotherhood.

As suggested, Abu Dhabi's campaign has three audiences; firstly, at a global level, by improving the image of Islam through public diplomacy, the country's rulers actually attempt to optimize their own image vis-à-vis several Western nations, notably amid their war in Yemen (2015–2019) (Al-Azami 2018). Secondly, at a regional level, the UAE fervently seek to discredit the Islamists, by accusing them of distorting 'true Islam'. Thirdly, domestically, the monarchy increases its immunity against demands for political reform.

## 2. Discussion

### 2.1. Contextualizing 'Moderate Islam'

The UAE post-Arab Spring policies may seem contradictory. On the one hand, regional interference triggered civil wars and humanitarian crises; the list is quite impressive: crushing Bahraini protesters, supporting the military putsch against the first democratically elected President of Egypt, bombarding Yemen together with the Saudis and imposing an aerial and naval blockade that resulted in the spread of famine and cholera, encouraging the secession of Aden in the country's south and, last but not least, militarily backing General Haftar against the UN-recognized government of Tripoli in Libya (Holmes 2014; Nour 2018; Ramani 2019; Abdulrahim 2019). With a similar attitude, Abu Dhabi introduced compulsory military service for 12 months in 2014 that was extended to 16 months in 2018 (El Yaakoubi 2018).

On the other hand, the government announced the creation of four new ministries—Tolerance, Happiness, Youth and Future—reflecting the country's 'success story' in spite of the upheavals surrounding it. In this framework, Dubai's ruler, Mohamed bin Rashid al-Maktoum, addressed the region and the world:

> The changes reflect what we have learned from events in our region over the past five years. In particular, we have learned that failure to respond effectively to the aspirations of young people, who represent more than half of the population in Arab countries, is like swimming against the tide ( . . . ) We do not forget that the genesis of the tension in our region, the events dubbed the "Arab Spring," was squarely rooted in the lack of opportunities for young people to achieve their dreams and ambitions ( . . . ) We have also learned from hundreds of thousands of dead and millions of refugees in our region that sectarian, ideological, cultural and religious bigotry only fuel the fires of rage. We cannot and will not allow this in our country ( . . . ) When the Arab world was tolerant and accepting of others, it led the world: From Baghdad to Damascus to Andalusia and farther afield, we provided beacons of science, knowledge, and civilization, because humane values were the basis of our relationships with all civilizations, cultures, and religions. Even when our ancestors left Andalusia, people of other faiths went with them. (UAE The Cabinet n.d.a)

Donor diplomacy followed suit:

The UAE aid has only humanitarian objectives. It is neither governed by politics nor is limited by geography, race, colour or religion of the beneficiary. This is a practical application of the principle of tolerance in the UAE. This policy was laid down by the founder President of the UAE, the late Sheikh Zayed bin Sultan Al Nahyan who stressed that foreign aid and assistance is one of the basic pillars of UAE foreign policy. (UAE Government n.d.)

The Emirates constantly boast about meeting the UN Millennium Development Goals in eradicating extreme poverty, while conferring pledges to alleviate the suffering of the Yemeni people. However, just pouring money into the same countries that they themselves have been shelling is not enough to win the hearts and minds of the international audience. That, of course, is not to say that Abu Dhabi bears the sole blame for the misfortunes of Yemen. Given their common fear of Iranian intentions in the wake of the Houthi takeover of the capital Sanaa in 2014, the UAE were invited by the Saudis to participate in their military campaign in order to restore the status quo ante. Whereas it is true that the Emirates were driven by their own regional aspirations, it would have been inconceivable for these two Gulf Arab powers to leave Iranian interference in the Peninsula unchecked. Despite the fact that Tehran denies it, a 2018 UN report indicated that the Houthis were supplied with military material emanating from the Islamic Republic (Fahim 2018).

In the midst of that new proxy conflict, the question of reputation, which is of crucial significance in public diplomacy, inevitably came to surface. Much like the USA during the Iraq War, whose crusade in exporting democracy was overshadowed by the inhumane treatment of prisoners in Abu Ghraib and Guantanamo (Nye 2019, p. 11), the Emirates are still haunted by the images of the famine-stricken children that went viral on social media during the Saudi-led Yemen campaign, in addition to UN allegations of committed war crimes and evidence of secret detention centers (Human Rights Watch 2018, World Report 2019). The Houthis also face war crime charges, such as diverting food aid or sending child soldiers to battle (BBC NEWS 2018; Human Rights Watch 2015). To make matters worse for the Emirati international image, the embattled Yemeni President, Mansour Hadi, accused the Crown Prince of Abu Dhabi, Mohammed bin Zayed of 'behaving like an occupier of Yemen, rather than its liberator' (Hearst 2017).

For this reason, pledges to the reconstruction of the war-torn countries and declarations on peace, tolerance and women empowerment in the Middle East and Africa (UAE The Cabinet n.d.b) must be viewed in tune with UAE efforts to create 'soft power', which is, according to Nye (2019, p. 11), the outcome of successful public diplomacy. By reinterpreting religious tradition on the basis of tolerance, openness, interfaith dialogue and moderation, the UAE do not try to sell, in Nye's words, 'an unpopular product' (p. 13), e.g., their policies in Yemen (Ghaith 2018), but rather to downplay the atrocities, the crackdowns on dissidents and the reported abuses of migrant domestic workers from India, Nepal, Indonesia, Sri Lanka, the Philippines, etc. (Human Rights Watch 2017). At the same time, the country improves its world profile and raises its credibility in international relations.

The ideological utilization of Islam, of course, is not a novelty. Both Islamic heritage and the Arabic language have traditionally predominated Arab cultural diplomacy to meet political ends; the Algerian National Liberation Front ruling party imported Azharite graduates from Egypt to reinforce the postcolonial state's Arab Islamic identity (Stora 2001, pp. 148, 171; Mansour 2005, pp. 275–81), Abd al-Naser justified nationalizations and land redistribution on religious grounds (Rahman 1982, pp. 88–89), whereas Saudi Arabia used the think-tanks of Salafists and self-exiled Islamists against Panarabism and Naserism and, later on, Iranian Shiite revolutionarism (Vassiliev 1998, pp. 469–72). Today, the international outcry provoked by the Islamic State in Iraq and Syria (ISIS) terror attacks in Europe, as well as, its heinous acts in the Middle East, such as beheadings and purges against religious groups, provided an opportunity to countries like the Emirates or Indonesia to redesign their public diplomacy by defending 'true Muslims' and 'true Islamic tradition' on behalf of the Arab and Islamic worlds (Hoesterey 2016).

The warm reception of 'moderate Islam' rhetoric lies in the historically cordial relations between the UAE and the West, which have been shaped by shared interests and security concerns. After the

British withdrawal from the Gulf, the country received for the first time American military material in 1977 (Sirriyeh 1984, p. 111); throughout the decades that followed the collapse of the Shah's regime, Emirati military buildup has reached unprecedented levels by virtue of the monarchy's geostrategic value in containing revolutionary Iran and al-Qaida alike (Young 2013). It is no wonder that the UAE wholeheartedly participated in the international coalition's fight against the Islamic State. In the meantime, their partners in the West turned a blind eye in respect to their acts in Yemen. After the Charlie Hebdo attacks, the Emirati foreign minister was among the first Muslim personalities who rushed to Paris to condemn the Jihadists and attend the rally of the world leaders (Tran 2015).

It was not until 2018 that some of the country's arms suppliers, like Germany, Norway, Austria, Denmark, Finland, the Netherlands, and, to a lesser extent, Sweden voiced their criticism over the war crimes that their Arab allies had allegedly committed in Yemen and imposed bans and restrictions on exports of military material (Westall 2019). Although the legal framework exists (EU Common Position on Arms Exports of 2008 and the Arms Trade Treaty of 2014), the position of the West towards that matter has been neither unanimous nor clear due to the lucrative contracts of the defense industries that lobby on behalf of their clients. The UK adopted a rather ambivalent stance, while French officials acknowledged that they have never ceased to provide Abu Dhabi and Riyadh with the latest weaponry in the name of fighting al-Qaida in the Arabian Peninsula (Mielcarek 2019). Hence, 'moderate Islam', in its own right, could be a useful public diplomacy tool in improving biased Western European perceptions of the Emirates.

As for the United States, they represent the UAE's major ally and strategic partner. After 9/11, they have been always receptive to their Muslim partners' 'moderate Islam' diplomatic initiatives, irrespective of human rights violations or their interventions in Yemen and elsewhere. Against the background of deteriorating US–Iranian relations in the post-Obama era, the circumstances have been favorable for the Emirates to project their model of 'moderate Islam' as a positive force capable of containing all forms of extremism. The various anti-terror pacts signed by the Trump administration and the Arab monarchs since 2017 set the geopolitical context for the Emirati 'moderate Islam' public diplomacy, which is examined at length in the lines that follow.

## 2.2. Putting UAE Public Diplomacy to Test

Taking into consideration the three dimensions of public diplomacy, i.e., news management, strategic communication and building long-term relationships with key individuals and populations (Leonard et al. 2002, p. 10), we could claim that all of them apply to the UAE 'moderate Islam' campaign, though with different degrees of success.

In the first place, even though the Emirati government tightly controls domestic media and promotes its peace and tolerance initiatives on Facebook and Twitter, it is impossible to distract regional and international attention from daily discussions about Yemen in light of the UN and Amnesty International reports implying Abu Dhabi's direct involvement in war crimes during the period 2016–2018 (Bruce 2018). For instance, to the accusations of unjustifiably killing thousands of civilians, including children, by targeting school buses, funerals and weddings, the UAE usually respond with ministerial tweets and press releases about the peaceful coexistence of the country's religious communities, sharing photos of newly erected churches or Hindu temples on their official social media pages. Despite scaling down their military presence since the summer of 2019, war information on Twitter continues and the UAE certainly lost most of its battles (Emirates Leaks 2019). On top of that, Emirati public diplomacy lacks a tool of the size and impact of Al-Jazeera that would be essential for shaping Arab and Western public opinion on current matters (Samuel-Azran 2013). This proved to be a major setback, especially since the Saudi–Emirati rift with Qatar in 2017. Al-Jazeera releases videos on its social media about Riyadh's and Abu Dhabi's role in the 'world's worst humanitarian crisis' (Gathmman 2019), whose negative impact cannot be easily reversed.

Abu Dhabi has been more skillful in creating 'soft power' as far as strategic communication is concerned. This second dimension of public diplomacy regards managing international perceptions

of the country as a whole through the mediation of several agencies and actors working in unison (Leonard et al. 2002, pp. 14–15); according to Nye (2019, p. 12), governments organize campaigns, like the 1980s peace movement diplomacy of the Soviets, that revolve around central themes and values and plan events such as conferences, partnerships, festivals, etc. to reinforce them. The goal of such a strategy is to affect international opinion in the long run. In recent years, the Gulf monarchy, indeed, has been quite active in its national rebranding along the lines of 'moderate Islam', as the Emirati Ambassador in the US, Yousef al-Otaiba, claimed in his CNN (Cable News Network) interview:

> Divisiveness and polarization are on the rise across the world, and—if left unchecked—this trend will undermine global stability and peace. The UAE is pushing against this rising tide by creating a model that can serve as a road map for others to promote greater tolerance and openness. Unique government policies, innovative partnerships and interfaith dialogues are three of the ways the UAE is leading by example. (Otaiba 2016)

The cornerstone of Abu Dhabi's strategic communication is the so-called 'Forum for Promoting Peace in Muslim Societies' (PEACEMS), which has been held five times since 2014. The first session's goals were outlined as follows:

(a)   Reviving the spirit of coexistence that used to preside in Muslim societies;
(b)   Reviving the humanistic values among all religions;
(c)   Resorting to scientific methodologies in order to correct distorted views on religion;
(d)   Encouraging the Ulema to preach tolerance and peace;
(e)   Enhancing the role of the United Arab Emirates in spreading peace, security and prosperity in Muslim and non-Muslim societies alike (PEACEMS 2014).

By referring to several Quranic verses (The Holy Quran 5:2; 8:1; 49:13; 2:269; 7:170; 16:90; 21:107; 39:10), the Forum defines the 'true Islamic' values, such as tolerance (al-tasāmuḥ), compassion (al-raḥma), reform of the self (iṣlāḥ al-dhāt), justice (al-'adl), patience (al-ṣabr), solidarity (al-taḍāmun) and religious centrism (al-wasatiyya). The UAE embassy in the US plays a vital role in rebranding Islamic values as the 'original' Emirati ones:

> The UAE has a new vision for the Middle East region—an alternative, future-oriented model that supports moderate Islam, empowers women, embraces diversity, encourages innovation and welcomes global engagement. These values have been ingrained in the UAE's DNA since the country's founding in 1971. It explains why over 200 nationalities call the UAE home and why different religions have built approximately 40 churches, two Hindu temples, a Sikh temple and a Buddhist temple, which welcome multi-national congregations. (Embassy of the United Arab Emirates 2019b)

Undoubtedly, international conferences constitute a powerful asset in public diplomacy. Together with the Moroccan dynasty, the Emirates provided their valuable 'know how' in holding the Conference for the Rights of Religious Minorities in Predominantly Muslim Lands in Marrakesh, in January 2016. The Conference resulted in the famous Marrakesh Declaration—the purpose of which is to alleviate Western concerns over the fate of non-Muslims in Muslim societies (Marrakesh Declaration 2016). Central themes in Islamic tradition and historiography such as the *Charter of Medina*—a document referring to Prophet Muhammad's contract with the people of Medina who offered him shelter after the Hijra (622 AD)—are reinvented under the prism of present-day minority issues, interfaith dialogues and common counter-extremism efforts. Another such event is the World Tolerance Summit that is hosted by Dubai every November, since 2018, in conjunction with UNESCO's International Day for Tolerance (16 November). Its first session was attended by 1866 participants from 105 countries around the world (Al-Shaibani 2019).

Strategic communication, though, is not limited to annual forums and conferences. Over the last five years, the government enacted a variety of laws and policies aiming to institutionalize the

concept of tolerance at home, while exporting the 'Emirati model' abroad. Just to mention a few, a Ministry of Tolerance has been established (Al-Zayani 2016); the National Program for Tolerance was launched in 2016, praising Emirati Islam and culture as 'truly moderate' (UAE The Cabinet n.d.c). In 2017, the—first of its kind in the Arab world—International Institute for Tolerance was founded by the country's PM, Sheikh al-Maktoum and, in late 2018, the Emirati President, Sheikh Khalifa al-Nahyan, announced the National Research Project on Tolerance, reminding the world that his country 'is now a global beacon of civilized behavior' (Khamis 2018).

The third dimension of public diplomacy is identified as relation building; this process takes place either through the mediation of key personalities (religious leaders, thinkers or politicians) or at the level of populations who work, study, participate in cultural exchange or simply visit the country under question. Just like Margaret Thatcher, Helmut Schmidt or Anwar Sadat, who, according to Nye (2019, p. 13), 'adopted and promoted American values, after having been educated in US institutions', the UAE seek to influence future Arab leaders by instructing them on their own tolerant values. The seminars and courses taught in the Mohammed Bin Rashid School of Government are indicative of Emirati intentions to exercise public diplomacy through educational programs (Mohammed Bin Rashid School of Government n.d.).

Furthermore, Abu Dhabi set up a special relationship with two prominent religious figures: Pope Francis and the Grand Imam of al-Azhar, Doctor Ahmad al-Tayeb. Sheikh al-Nahyan visited Pope Francis in 2016 and then the Pontiff repaid the visit in February 2019. During his historic speech in front of more than 150,000 people, the Pope commented on the war in Syria and Yemen in a rather ambiguous way; the serious accusations concerning Abu Dhabi's deep indulgence in the humanitarian crisis of Yemen did not prevent Pope Francis from whitewashing his hosts' alleged war crimes, when he stated that 'the UAE are a modern country and an open society to dialogue that educate its children by looking forward' (Barbarani 2019). Concerning the Grand Imam of al-Azhar, he was singled out thanks to his influential status in the Sunni world; he had already met Pope Francis at the latter's visit to al-Azhar in 2017 and all Muslim autocrats consider him an indispensable interlocutor. During the Pope's UAE visit, he was invited to cosign the 'Human Fraternity Document' with the Pontiff (Associated Press 2019). Consequently, the UAE capitalized on the presence of those high-profile personalities to create 'soft power' via interfaith dialogue diplomacy.

At the level of peoples' public diplomacy, Leonard stresses the need 'not just to develop relationships but to ensure that the experiences which people take away are positive and that there is follow-up afterwards' (Leonard et al. 2002, p. 18). This can be true for a country like the Emirates, where more than 80 percent of the country's population according to 2019 data consists of Indian, South Asian, European and other expatriates (United Arab Emirates Population Statistics 2019). Constructing new places for worship and protecting the rights of every faith are adequate reasons to keep non-Muslims content. In 2017, in a strongly symbolic and rather spectacular gesture, Sheikh Mohammad bin Zayed Mosque in Abu Dhabi was renamed to Mariam Umm Eisa, honoring the mother of Jesus. Hence, it is no wonder that American pastors publish articles praising the tolerant, multireligious Emirati society:

> Different religions have built 40 churches, two Hindu temples and a Sikh temple, all on land donated freely by the ruling authorities, who welcome multi-national congregations in the UAE. The region's largest Anglican Church is currently being built in Abu Dhabi and will accommodate more than 4000 worshipers once complete. (Reverend et al. 2018)

*2.3. Containing the Threat of Political Islam*

In this chapter, it is examined how 'moderate Islam' serves as a tool of 'sharp power' in terms of containing the perceived or real threat of Political Islam. Reading between the lines of the abovementioned interfaith partnerships, as well as the declarations on tolerance, moderation and peaceful coexistence, it is argued that Abu Dhabi's and Dubai's rulers do not consider the fanatics of ISIS or al-Qaida per se to be their real enemies. Taking into account the UAE socio-political structures, the actual threat emanates from activists who not only ask for more liberties but also question the

royal absolutism. As the modern history of the Arab world has shown us, political dissidence in these countries is predominantly Islamist[1] in nature. During the third annual session of the PEACEMS forum in 2016, participants confirmed that:

> Our greatest concern is to shed light on the political outcomes that stem from the distortion of the Islamic Sharia ( . . . ) interpreting the teachings of our religion out of their original context has given the necessary pretexts to forces threatening social peace in Muslim societies, specifically those who are still trying to recover from regime change, meanwhile other forces have even attempted to reshape international order and replace the nation state with their self–declared 'Islamic Caliphate'. There is no parallel in these entities' usurpation of religious symbols and terminology. (PEACEMS 2016)

The previous passage directly refers to the Islamic State in Iraq and Syria in addition to Egypt's Muslim Brotherhood experience, castigating the subversive role played by the Islamists every time they have meddled with politics. What is even more interesting, though, is how the Emirati 'moderate Islam' campaign demonizes Islamists of all stripes. In other words, under the pretext of 'distorting Sharia and the teachings of religious tradition', non-militant Islamist parties committed to democracy, elections and socio-political reform are easily equated with Osama bin Laden, the Taliban or Abu Bakr al-Baghdadi. As one Emirati writer puts it 'Islam has been hijacked by the Islamists and should be reclaimed by the forces of moderation' (Al-Sawafi 2014). The whole campaign is reminiscent of the post-9/11 USA neoconservative 'bad and good Muslim' rhetoric.

Regionally, this anti-Islamist strategy has a twofold target, bearing ideological, religious and geopolitical implications: disparaging the well-organized, transnational Muslim Brotherhood (al-Ikhwan), while counterbalancing Qatar—another regional player whose small size did not prevent it from resorting to 'hard power' diplomacy after 2011 (Nuruzzaman 2015, pp. 226–29). In order to put the UAE 'ikhwanphobia' into context, we should first have a brief look at the Islamist movement's political evolution in the Middle East and North Africa (MENA) and the Gulf in particular.

Most of the times during its turbulent history, the Brotherhood was either tolerated by the post-colonial state or persecuted, depending on the domestic situation, on the reevaluated regional policies of the Arab countries and on their shifting alliances. During the 1970s and the 1980s, some splinter Brotherhood groups, like al-Jamā 'a al-Islamiyya, elaborated the idea of Takfīr (excommunication of Muslims) and waged Jihad against the Kuffār, i.e., the perceived 'infidels' who ought to be fought and killed either at home or abroad (Keppel 1985, pp. 71–91, 194–210; Pargeter 2010, pp. 181–85). The emergence of such extreme currents has been a mixed blessing; on the one side, it enabled the 'loyal' Brotherhood leaders to redefine themselves as 'moderates', while whitewashing their own socio-political conservatism and illiberal stance on issues like the segregation of sexes. On the other, this very fact justified harsh state suppression of Islamism sui generis. With a few exceptions like Syria, where the Brothers tried to topple the government by force in 1982, in other countries like Egypt, Jordan, Morocco, Kuwait or Yemen, they remained committed to charity and preaching (Clark 2004, pp. 42–145). During the last 30 years, the would-be 'moderate Islamists' formed

---

[1]　As Islamists (al-islamiyeen) are defined those Muslims who consider Islam to be a complete socio-political and economic system, as well as, a cultural program, rather than a religion that is concentrated solely on spiritual matters. For this reason, analysts tend to use the term Political Islam and Islamism interchangeably. The historical matrix of this modern ideology was the Egyptian Muslim Brotherhood and the writings of its founder, the school teacher Hasan al-Banna (1906–1949). The evolution of Islamism has gone part and parcel with the political and social development of the Arab and the Muslim nation-states throughout the 20th century. Some groups of Islamists turned out to be extremely militant, engaged in terrorist attacks against Muslims and non-Muslims alike and even attempted to overthrow governments via armed revolts, whereas other groups respected central authorities and generally espoused non-violence. In terms of European political standards, those Islamists could be described as socially conservative and economically liberal, i.e., the 'Islamic equivalent' of Christian democratic center-right parties. An Islamist party (Al-Nahda) has been part of the Tunisian governing coalition for many years, while AKP, Erdogan's ruling party, is also considered Islamist in its ideology and origins. In Egypt, although the Muslim Brothers won parliamentary and presidential elections in 2012, they demonstrated illiberal tendencies in order to secure their rule.

parties and even elected their own deputies in national parliaments or participated in governments (Anani 2010, pp. 1–6).

Undeniably, the watershed in the organization's political fortunes was the Arab Spring. In the aftermath of the Tahrir Square uprisings, Egyptian Islamists won two consecutive elections, both parliamentary and presidential. The association founded by a humble school teacher 84 years ago had become the protagonist of the most pivotal Arab country's political transition. Islamist rule in Egypt was a reality, albeit a short-lived one. Mass protests, new alliances on the streets among those who were sidelined by the Brotherhood, not to mention relentless foreign interference, boiled down to Morsi's downfall. The Islamist cabinet and President Morsi were accused of incompetence, favoritism and autocratic tendencies; these claims are verified to some extent, considering the illiberal and socially conservative constitution that had been brought to popular vote (Hamid and Wheeler 2014). The country was at the brink of civil war when Abdel Fattah al-Sisi's 2013 3rd July coup took place, allegedly to 'save the 25th January revolution from the Islamists who hijacked it' (Kourgiotis 2014).

The sudden perish of the Muslim Brotherhood in its own birthplace echoed the post-Arab Spring contest for power among the Gulf Cooperation Council countries, which were alarmed by the Islamists' electoral victories and bitterly divided over their reactions towards them. We should keep in mind that by the time of the Arab uprisings, Brotherhood-affiliated Islamists had already established their presence in the Gulf countries dating back to the first purges under Naser (1952–1970). Thousands of Brotherhood members contributed in the social and educational development of the Arab Gulf and, as a matter of fact, they infiltrated those countries' nascent universities. The Emirati Brotherhood particularly had grown so powerful in Abu Dhabi's only university that, by the late 1980s, it was in a position to approve or reject the federal government employees' applications to the scholarship committee (Al-Rashid 2013).

Over all these years, relations between the Brothers and their wealthy hosts in the Gulf were based on a simple 'social contract': shelter for da 'wa (preaching) in exchange for refraining from politics. Younger generations, nonetheless, were influenced by the Brotherhood's ideological reorientations and started talking about political reform even inside the less liberal kingdoms. That is the case of the 1990s al-Sahwa al-Islamiyya (Islamic Awakening) in Saudi Arabia, as well as Da 'wat al-Iṣlāh (Call for Reform), or simply al-Iṣlāh, in the Emirates. Not surprisingly, after starting monitoring Ikhwani activity in the universities and elsewhere, the UAE authorities demanded from al-Iṣlāh to cut its organic ties to the mother organization in Cairo (Freer 2015a, pp. 11–3, 18–20; Al-Rasheed 2002, pp. 176–83; Lacroix 2011, pp. 37–80). The Emirates had enough good reasons to feel threatened by the Brothers, because they left their activities unchecked for two decades at least.

On the contrary, the al-Thani dynasty of Qatar felt no existential threat to its rule by the Brotherhood influences in this tiny kingdom, because as Grabowski states 'Emir Hamad al-Thani left no space for the Brothers to gain support through social services, creating jobs etc.' (Grabowski 2016, p. 358). The Emir Tamim bin Hamad al-Thani and his father before him, in that movement's capabilities to fill the vacuum left by the demise of the Arab autocrats, they saw a golden opportunity to maximize Qatar's regional standing (Roberts 2014). Erdogan's government acted in full conformity with al-Thani dynasty in creating a chain of 'loyalist Islamist republics' across the Arab world on behalf of 'the oppressed people' as the Turkish President has declared publicly (Daily 2014). Accordingly, Egypt, during the short term of Morsi's rule, received eight billion US dollars from Doha (Kerr 2013). In the same vein, Qatar extended its support to the Tunisian Islamist party Ennahda that dominated the 2011 constituent assembly following the country's first free elections and kept backing financially the post-2014 coalition government, much to the dismay of Abu Dhabi (Cherif 2017).

The Saudis and the Emiratis feared that, had the Ankara–Cairo–Doha axis remained intact, it would have probably emboldened the rest of the region's Islamists, including their own, to demand reforms, elections, etc. (Fajri 2013). As soon as Hosni Mubarak was deposed, Hizb al-Ummah al-Islami, the first political party in Saudi Arabia, had already been banned after a very brief 'Saudi Spring' (The Islamic Ummah Party n.d.). As for the UAE, anti-ikhwanism reached new heights; as early as

2011, members of al-Iṣlāḥ joined forces with liberal non-Islamists and signed a petition requiring universal suffrage, legislative authorities for the Federal National Council[2] and broader constitutional reforms (Freer 2015b). A year later, Emirati authorities discovered an alleged coup attempt. According to Emirati sources, 60 Brotherhood affiliates were arrested and charged with plotting against the federal government (The National 2012). Al-Nahyan ruling dynasty referred to an 'international Brotherhood conspiracy coordinated by neighboring countries', although President Morsi had repeatedly reassured the Arab leaders that 'Egypt had no intention of exporting its revolution' (Saleh and Hall 2012).

This Emirati version of 'McCarthyism' has led to the diplomatic standoff with Qatar, crises with the UK and other European countries due to their reluctance to designate the Brotherhood as a terrorist group and even to a severance of relations with their protégé, the Yemeni President, because of his ties with the local Ikhwanis, namely the al-Islah party (Cafiero 2018). It is worth mentioning that some UAE based academics from Europe or the US try to apologize for their hosts' 'ikhwanphobia', arguing that the Brotherhood's ideology runs counter to the country's 'moderate Islamic' model. They reiterate that authorities have clamped down only on the Brotherhood-affiliated activists and that non-Islamist dissidents who call for reforms are generally tolerated (Forstenlechner et al. 2012). Such claims are rather dubious. In the meantime, the UAE government acknowledged that the 'Islamist threat' has been graver in the poorer northern emirates and pledged to abolish inequalities in the distribution of the wealth, implementing the goals set by the Dubai 2021 Plan (Moshashai 2018, pp. 21–22).

Apparently, Saudi Arabia and the UAE had a strong interest in interfering with Egypt in the wake of the 2013 anti-Brotherhood protests. According to leaked Saudi cables, the two Gulf partners favored what has been known as the 'Pakistani model' (Royal Embassy of Saudi Arabia in Cairo 2012), i.e., supporting military strongmen and cultivating relations with local religious authorities. This is exactly the same counter-revolutionary scenario that they tried to impose in Libya through General Haftar's offensive against Tripoli's Islamists. Regardless of investing billions of dollars so as to outflank Qatari influences and help Sisi stabilize his regime (Farouk 2014, pp. 10–13), 'healing' the Muslim Brotherhood 'disease' is not just a question of money; an effective 'medicine' had to be found, combining coercion, aid policies and—most importantly—ideological 'rehabilitation'.

Whereas the Saudis invested in their loyalist Salafists of Hizb al-Nour as a means to avert the pious masses from the Ikhwan (Kourgiotis 2016, pp. 13–21), the UAE took advantage of Egyptian 'recovery' from Political Islam to initiate their 'moderate Islam' policies. The contribution of al-Azhar University can be considered critical to the common cause of 'reclaiming Islam' and containing the Ikhwanis whether in Egypt or the Gulf. Historically, in the Sunni lands, the Ulema intervened in the production of religious tradition at the behest of central authority, no matter whether they were the Caliphs, the colonial empires or today's nation states. At the request of the UAE rulers, Islamists were portrayed by the Ulema as 'sick people' who needed 'immediate treatment':

> The forum's biggest concern revolves around how religious fanaticism and terrorism will be defeated in the hearts of the people before materializing into actions ( … ) it is the duty of the Ulema, the governments and other institutions to instruct Muslims on the proper forms of religiosity and religious behavior ( … ) there is no doubt that our Islamic pharmacy is able to prescribe the right medication to the patients in order to get rid of the disease of violence once and for all. (PEACEMS 2014)

In the passage of the last five years, the Emirati–Azharite partnership came to the surface as the bastion of 'moderate Islam'; an alternative to both Ikhwanism and Salafism. The Egyptian President, Sisi, has

---

2    The UAE parliament consists of 40 members—20 of them are elected every 8 years, while the rest are directly appointed by the Ruler's Court in every Emirate. Its authority is mostly consultative and not legislative, given the fact that it has the right only to review the laws presented to it by the Cabinet. No federal law can pass without the final approval of the Emirati Cabinet. Available online: https://www.government.ae/en/about-the-uae/the-uae-government/the-federal-national-council- (accessed on 16/11/2019).

been cautious enough to undertake a national campaign of fighting atheism, while pressing al-Azhar to revise its curricula in the direction of al-wasatiyya (religious centrism) and apply scrutiny on suspicious fatwas (Ibrahim 2014; Mourad and Bayoumi 2015). After the ordeal in Bataclan, Al-Azhar even asked the French government to send its imams to teach French Muslims 'moderate Islam' (France24 2015). In the 2016 Grozny conference, the UAE together with al-Azhar and Russia redefined what Sunni Islam 'ought to be' and irritated the Saudi Ulema by excluding Salafism as intolerant and prone to Takfīr. The invitation of the Grand Mufti of Syria was a sign of the Emirati–Egyptian intentions to co-opt another staunch enemy of the Ikhwan, Bashar al-Assad (Diwan 2016). The Emirates for their part, organized a series of lectures delivered in the mosques of Dubai, throughout 2019, highlighting Islam's moderate stance on scientific, social and religious topics (Gulf News 2019).

　　From every aspect, 'religious rehabilitation' supplements the politics of containment. All those years of instructing Muslims on how to 'properly express their religiosity' has taken place in parallel to the Emirati airstrikes against the 'Caliphate' in Syria and Sisi's efforts to eliminate the Jihadist insurgency that his regime was constantly facing in the Sinai Peninsula from the very beginning. The UAE leadership intensified its security cooperation with the new Egyptian government and, at the same time, it took preemptive measures at home, such as the Anti-Terrorist Crimes Laws No. (7) and No. (9)/2014, the Anti-Hatred and Anti-Discrimination Law No. (2)/2015 and the Tolerance Law No. (9)/2017 (UAE The Cabinet n.d.d; Hamdy 2018). Moreover, two counter-terrorism centers, namely al-Hedayah (Guidance) and al-Sawab (Right), were established to eradicate Da 'esh ideological imprints on the Muslim youth (Sky News Arabiyya 2015).

　　Once again, the Jihadist threat proved to be a very convenient alibi; in terms of internal security, it justified the expansion of the UAE anti-Islamist crusade so as to include any kind of opposition. For one thing, Abu Dhabi's first PEACEMS gathering of Sunni Ulema from around the globe to discuss de-radicalization coincided with the public circulation of a list of 85 terrorist organizations including al-Iṣlāh and all the regional branches of the Muslim Brotherhood. The Emirates went as far as to target legally registered Islamic Associations and NGOs (Non-Governmental Organizations) tolerated by the US and European authorities on the grounds of disseminating dangerous Ikhwani ideology (Gulf News 2014). In the same year, the Council of Muslim Elders was founded in Abu Dhabi and the Grand Imam of al-Azhar was appointed as its chairman (Muslim Council of Elders n.d.). The Council functions as a counterweight to the Doha-based International Union for Muslim Scholars headed by the Brotherhood Imam and al-Jazeera TV star, Yussef al-Qaradawi (International Union for Muslim Scholars n.d.). Fierce theological battles have been fought between the two bodies ever since. Time will tell whether the UAE will remain the staunchest anti-Brotherhood power in the region even in light of a future rapprochement with Qatar.

　　Despite 'moderate Islam's' bias towards Salafists, the two models share the same quietist approach, i.e., de-politicize Islam as a way of de-radicalizing Muslim youth; in their view, the tolerant Muslim should be the apolitical ones. Similar initiatives were launched after the 9/11 terror attacks, like the Commonwealth Office's program 'Projecting British Islam'. Even if they failed to prevent the 'Hijrah' of European Jihadists, such programs have always been appealing to the insecure Arab dictators (Mandaville and Hamid 2018, pp. 2–8). By 'reclaiming Islam' from the Islamists, the state power reasserts its monopoly on the use of religious symbols, discourses and Quranic interpretations; Jihad and the Sharia are incorporated in the policies of the nation state. What is more, several regimes shield their immunity to potential political change.

　　In conclusion, from 2014, the UAE underwent a new nation-building process that illuminated the necessity to 'instill tolerance in the Emirati national consciousness' and 'forge future tolerant generations among the country's youth' (UAE Ministry of Tolerance n.d.). In that sense, 'true Islamic values' are restored for the society's sake, whilst Islamist and non-Islamist dissidents alike are ostracized as 'the enemies of religion and the motherland'.

### 3. Conclusions: Caught between 'Soft' and 'Sharp Power'?

This essay presented a detailed account of how the top–down project of 'moderate Islam' has redefined the UAE's international image as well as their post-Arab Spring regional policies. Time and again, colonial and postcolonial authority did not resist revisiting religious tradition under the pressure of external or domestic challenges. The Emirates' policies fit that historical pattern. The article concludes that, via their campaign, the UAE engaged with 'soft' and 'sharp power' alike. Throughout our presentation, we analyzed how religious tolerance has been utilized to whitewash domestic political intolerance and aggressive interventionism in other countries' internal affairs.

With respect to public diplomacy, the joint de-radicalization efforts between the Emirates and al-Azhar are deemed as exemplary tactics in creating 'soft power'. Public opinion, of course, has not forgotten Yemen, or the Emirati occupation of some parts of this country, yet the UAE leadership managed to rebrand itself as one of the world's earnest backers of interfaith dialogue and redirect the agenda to its favor. Pope Francis' visit stands as a testimony to the success of such diplomacy. Moreover, thanks to 'moderate Islam', the small Gulf monarchy re-emerged in international relations as a model exporter; following Abu Dhabi's lead, even the Saudi Crown Prince, Muhammad Ibn Salman, appears determined to reintroduce 'moderate Islam' in his ultra-conservative kingdom, in line with Riyadh's Vision 2030 (Kingdom of Saudi Arabia n.d.).

However, due to growing security concerns, the Emirates had to resort to what Nye calls 'sharp power' (Nye 2019, p. 17), such as funding anti-Brotherhood parties and militias, backing coups and disseminating conspiracy theories to discredit opponents. These practices are identical to US anti-communist activities during the Cold War and on no account produce 'soft power'. Perhaps, the UAE could be better compared to the Chinese model that rests on 'soft power' campaigns, e.g., ecological issues, climate change, etc., whilst crushing dissidents and ethnic groups in the interior (Nye 2019, pp. 18–19). Against the backdrop of increased calls for reforms, whether social, economic and, to a lesser extent, political, 'moderate Islam' endorses and legitimizes the status of the UAE monarchy by casting out opposition as 'intolerant' and consequently, 'unislamic'. Emirati policies will continue to be caught between 'soft' and 'sharp power' owing to authoritarian tendencies, unless bold constitutional reforms are instigated under mounting social pressure. This is a potentiality that does not fit into the monarchy's 'success story' for the time being.

Meanwhile, the monarchy is represented in the eyes of the US and some other Western partners as an 'enlightened model' for the rest of the region, despite the debates that have sparked in the parliaments of the UK, France, Germany, Sweden and elsewhere regarding the situation in Yemen. According to the Dubai 2021 Plan, the Abu Dhabi 2030 Vision and many others (UAE The Cabinet n.d.e), the ruling families aspire to transform the Emirates into a global hub of market economy, green growth and tolerance. This is the climax of the UAE rulers' public diplomacy indeed, notwithstanding that they could hardly tolerate political openness out of fear that it would backfire and cost them their centuries-old grip on power. As far as their Arab and Muslim audiences are concerned, the Emirati rulers are cautious enough to remind them that far from assimilating 'alien, Western values', they simply rediscover the Islamic Golden Age (9th–12th centuries) and revive the legacy of Damascus, Baghdad and al-Andalus.

**Funding:** This research received no external funding.

**Conflicts of Interest:** The author declares no conflict of interest.

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
