# Peer review of "‘Moderate Islam’ Made in the United Arab Emirates: Public Diplomacy and the Politics of Containment"

_religions, doi:10.3390/rel11010043_

Round 1

Reviewer 1 Report

This is an interesting and informative paper that has some good content which will be of interest to researchers in the field. 

It does however, require some improvements in language, grammar and tone as some of it reads more like a journalistic article than an academic paper. For example in lines 27, 132, 235 and 314 need better phrasing.

The conclusion also needs developing as it currently is rather thin and should more fully restate the papers overall argument and any additional reflections.

Author Response

Dear reviewer,

due to some additions requested by the second reviewer, the prhrases you mentioned can be now found in:

line 27, 151, 276 and 360

as for the conclusion, it is now merged with the former discussion in order to be wider, more analytical and representative of the essay's arguments with a minor addition in lines 529 - 531.

Please accept my kindest regards,

wish you a happy new year

Reviewer 2 Report

This article addresses a very timely topic: the use of the idea of “moderate Islam” as an asset in diplomacy and as an excuse to crush political dissent. This tendency can be seen in numerous countries from Central Asia to Russia and from Caucasus to North Africa. Therefore, this article is an important contribution to the study of Islam and politics as well as Islam and international relations. The author uses multiple different sources and clearly is very familiar with the Middle East politics. However, the article could be written in a more balanced way. In addition, the structure of the article could be a more logical. Therefore, I recommend publishing this article only after minor revisions.

The stated aim of this article is to analyse the use of the “tolerant” or “moderate Islam” in internal and foreign politics by using the United Arab Emirates (UAE) as a case study. This requires explaining how the concept is used to whitewash questionable policies in abroad and in the country. As rightful as the accusations on the violations of human rights against the UAE are, the author should be careful that the article does not begin to seem moore as a politically biased attack on the country. Two ways to avoid this are not downplaying similar violations in the opponents of the UAE and using language as neutral and precise as possible. When the author claims that the UAE cracks down pro-democratic dissent under the auspice of the fight with intolerant Islam, the point of reference is mostly Muslim Brotherhood (MB). The argument is well-grounded, but the discussion about the MB is occasionally rather uncritical. The discussion would seem more unbiased (and thus more convincing) if the author would note that the political line and the activity of the MB also contains illiberal features that set restrictions and may narrow down the idea of democracy.

While the argumentation of this article is convincing, the structure of the article is not always easy to follow. The subtitles do not always portray the content faithfully. For example, “reclaiming Islam from Islamists” is also discussed in the previous subchapter and the same or at least similar claims are repeated in different parts. The generic subtitle “results” does not tell much about the content. The discussion as such is well-written, so my only suggestion concerns restructuring it and adding some new subtitles (eg. when the discussion about three dimensions of public policy begins).

Some central concepts could be defined more clearly. How does the author define “Islamism” in or “Islamist” in this article, who or what groups, ideologies or religious traditions can be counted as such? Is soft power a goal of politics or can it also be means to attain some goals?

The author discusses the war crimes that the UAE has committed in abroad and the crack-down of opposition in the country. However, when the religious tolerance as a way to make the country appealing to foreigners is discussed, the author might wish to mention that there are also low-payed migrant workers, whose human rights are often trampled.

I realize that the reception of the diplomacy of the UAE is not the topic of his article. However, it would be good to say few words about situation in the international relations that explains this. While many countries that commit serious violations of human rights aim to whitewash these with the rhetoric of “moderate Islam”, they may not be as successful as the UAE. I would argue that the reason for this is not necessarily in the skillfulness of this rhetoric, but in the international situation. The UAE and Saudi Arabia are important allies for the West and especially for the USA in the difficult situation in the area. Moreover, the UAE is an opponent of Iran, one of the main “enemies” of the US in recent years. Yet, as the author notes, the war in Yemen has made some West-European countries more critical of Saudi Arabia and the UAE. However, in the last paragraph the author claims that “the monarchy is represented in the eyes of the West as an enlightened model” as if there would be some kind of unanimous “West”.

The text contains many parts, where the author explains “(italics mine)”. However, technically the italicized parts do not seem to be quotations as one would expect. Usually there is no need to emphasize important parts or concepts with italics and definitely no need to explain that the author has done it herself. On the last page, the “UAE based academics from Europe and the US” should be rephrased “some UAE based academics…. (Also, the name of the first author in the next reference is Forstenlechner, not Forstelnecher.)

Author Response

Dear reviewer,

thank you for your remarks, i tried to do my best 

this is my point - by - point response to your comments

second paragraph: 

i realize the question of avoiding bias, so i made the following changes/additions in respect to UAE opponents in Yemen and the Muslim Brotherhood

lines 104 - 122 were added about Houthi - Iran and alleged war crimes committed by all sides (not only the UAE or the Saudis)

line 307: a footnote that mentions Islamists' social conservatism in general and illiberal tendencies while in power in Egypt.

line 340 - 342: a more critical mention of the moderate Islamists

line 353 - 357: reference to the negative features of Morsi's rule

line 378 - 379: explaining why UAE feel threatened by the Brothers

third paragaph:

the title results has been removed and renamed to discussion; three new subtitles dividing it into three chapters: 2.1 Contextualizing 'Moderate Islam' 2.2. Putting UAE public diplomacy to test (no changes here) 2.3. Containing the threat of Political Islam

the discussion has now merged with the conclusion (lines 484 - 536)

fourth paragraph:

Islamism is now defined as a term in a footnote (line 307)

fifth paragraph:

the abuses of migrant domestic workers are now mentioned (line 130 - 131)

sixth paragraph:

the question of reception of Moderate Islam, as well as, the international context are now added and analyzed to some extent as requested (lines: 144 - 149, 158 - 176 & 528 - 531)

seventh paragraph:

italics mine ommited

Forstenlechner OK

Wish you a happy new year!
